# Wind Speed Measurement by an Inexpensive and Lightweight Thermal Anemometer on a Small UAV

**Jun Inoue** [1,2,*] and **Kazutoshi Sato** [3]

1    National Institute of Polar Research, Tachikawa 190-8518, Japan
2    Department of Polar Science, The Graduate University for Advanced Studies (SOKENDAI), Tachikawa 190-8518, Japan
3    Kitami Institute of Technology, Kitami 090-8507, Japan
*    Correspondence: inoue.jun@nipr.ac.jp

**Abstract:** Profiling wind information when using a small unmanned aerial vehicle (sUAV) is vital for atmospheric profiling and monitoring attitude during flight. Wind speed on an sUAV can be measured directly using ultrasonic anemometers or by calculating its attitude control information. The former method requires a relatively large payload for an onboard ultrasonic anemometer, while the latter requires real-time flight log data access, which depends on the UAV manufacturers. This study proposes the feasibility of a small thermal anemometer to measure wind speeds inexpensively using a small commercial quadcopter (DJI Mavic2: M2). A laboratory experiment demonstrated that the horizontal wind speed bias increased linearly with ascending sUAV speed. A smoke experiment during hovering revealed the downward wind bias (1.2 m s$^{-1}$) at a 12-cm height above the M2 body. Field experiments in the ice-covered ocean demonstrated that the corrected wind speed agreed closely with the shipboard wind data observed by a calibrated ultrasonic anemometer. A dual-mount system comprising thermal anemometers was proposed to measure wind speed and direction.

**Keywords:** wind speed measurements; laboratory experiments; field experiments in a cold region

## 1. Introduction

The meteorological use of unmanned aerial vehicles (UAVs) has attracted researchers and weather operational centers [1]. Fixed-wing UAVs have already been applied to operational numerical weather predictions (NWPs) for tropical cyclones [2]. Over the Arctic Ocean, where accessibility is low, the monitoring of atmosphere, ocean, and sea-ice have been replaced by fixed-wing UAVs [3–6]. This system requires a runway and special operators; logistical and human resources limit their wider use. The more systematic, straightforward operation of small UAVs (sUAVs) [7], such as rotary-wing UAVs, attracts people interested in using them. Significantly, the World Meteorological Organization (WMO) has held several meetings on the use of UAVs for operational meteorology [8] and plans its Demonstration Campaign in 2024 (https://community.wmo.int/uas-demonstration (accessed on 1 October 2022)).

The impact of UAV profiles on the NWP has been investigated, targeting many places as case studies. Flagg et al. [9] demonstrated that the observing system experiments consistently revealed a substantial bias reduction in predicted temperature and moisture profiles throughout the campaign period because of the assimilation of the fixed-wing UAV observations. Such a result was also found in a study targeting the complex terrain region [10]. Although radiosondes are fundamental observational systems for obtaining all tropospheric states, extra launches for operational purposes are costly, especially in remote regions such as polar regions and developing countries that cannot maintain permanent upper weather stations.

The ideal heights for UAV operation and optimum observing density of UAVs to improve the skills of NWPs were also examined using an observation system simulation

experiment approach [11]. High-quality atmospheric boundary layer observations by sUAVs have been conducted to fill the current operational observation gaps for NWPs and satisfy the requirements of WMO's Observing Systems Capability Analysis and Review tool (OSCAR) [12]. However, many issues remain regarding sUAVs.

Among the most significant issues in measuring atmospheric parameters using sUAVs is high temperature biases caused by the heat from the sUAV body and rotors and the heat directed on the thermistor from solar radiation, which also causes dry bias in relative humidity. The relationships between hot spots of an sUAV and temperature sensor placement have been investigated with laboratory experiments, especially by considering the rotor down-wash [13–15]. Several field campaigns demonstrated the effect of solar radiation on warm biases of approximately 0.6 °C [16,17]. The UAV intercomparison project demonstrated the benefit of aspiration of the sensors by rotor down-wash [18]. With appropriate sensor treatments, the quality of temperature data is comparable to the commercial meteorological sUAVs [15].

There are several choices for obtaining wind data. First, without an additional payload, wind speed and direction can be estimated based on sUAV dynamics (e.g., roll and yaw angle) [19,20]. This method is ideal because it does not require extra sensors. However, the algorithm is independent of each sUAV type, and several parameters must be adjusted for extra payload (e.g., a case of loading other sensors). More importantly, real-time detailed flight information from the commercial sUAVs is rarely obtained. This scenario becomes critical in flights of long-distance or very high altitudes beyond the visual line of sight (BVLOS) control. Real-time wind data transfer to a land station is desired.

Second, an additional wind sensor (e.g., an ultrasonic anemometer) can be applied to an sUAV. Barbieri et al. [18] demonstrated that ultrasonic anemometers on hovering sUAVs provided the most consistent results compared with other wind estimations. The challenge for measuring the wind profiles by ultrasonic anemometers is to reduce the effect of ascending and descending motions of the sUAV on the horizontal wind because measuring winds by hovering at many flight levels is not practical for the limited flight endurance of the onboard battery. The airflow near the anemometer during vertical movement is modified by the shape of the sensor and sUAV body. Furthermore, the weight of ultrasonic anemometers (typically 50–100 g), including a mounting pole on the top of an sUAV and a data processing device with a battery, limits the size of the sUAV.

This study aims to apply a thermal anemometer for wind measurement. Although a thermal anemometer is unsuitable for flights into clouds and under precipitation, the typical flight scenario does not experience such conditions. Lightweight and inexpensive features might expand the opportunities to measure winds by commercial sUAVs. In this study, laboratory and field experiments were conducted to investigate the feasibility of a thermal anemometer for wind profiling by an sUAV.

## 2. Materials and Methods

### 2.1. A Thermal Anemometer: HWS-19-ONE

HWS-19-ONE (Hortplan LLC, Japan) (hereafter, HWS) is a thermal anemometer system with two wind probes on the front and back sides, enabling a rough estimate of the wind direction (front wind or rear wind) (Figure 1a). Its specifications are summarized in Table 1. This system (1) is very lightweight (1 g), (2) has a wide measuring range (0–20 m s$^{-1}$), (3) has a frequent measuring interval (0.25 s), and (4) has minimal directivity (±10 %), which is very suitable for wind speed measurements by a commercial sUAV.

The manufacturer examined the sensitivity of wind speed to azimuth angle (i.e., directivity) in detail (Hortplan, LCC, Japan). A test by changing the wind direction by 10° around the sensor under 5 m s$^{-1}$ horizontal wind condition revealed that the observed wind speed only varied ±10% (the figure is available at https://www.sg-lab.info/wp_sglab/wp-content/uploads/2021/04/image01-1.jpg (accessed on 1 October 2022)). This low directivity indicates that the HWS sensor is a high-quality thermal wind sensor.

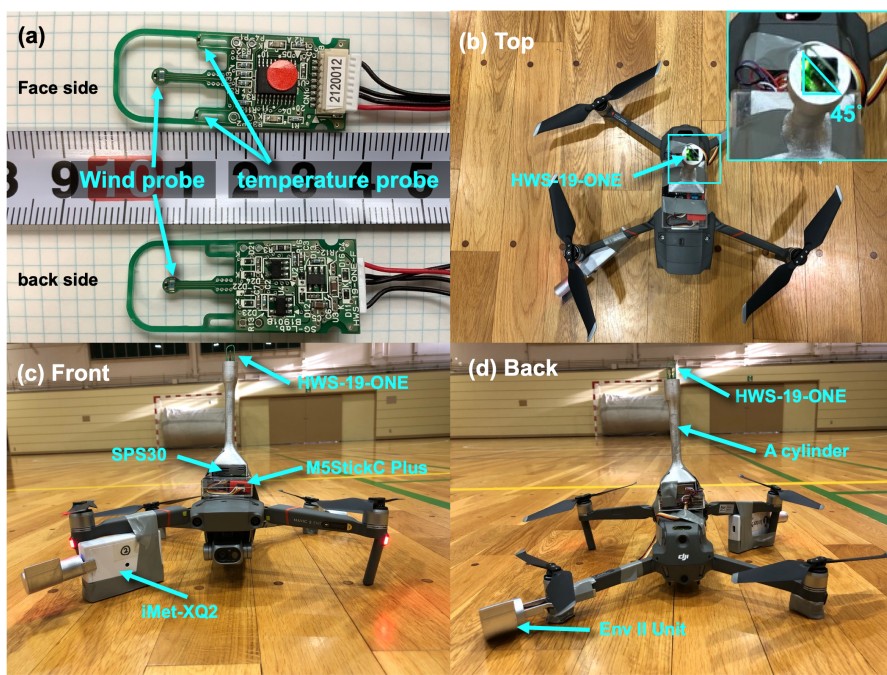

**Figure 1.** (**a**) HWS-19-ONE, (**b**) top view, (**c**) front view, and (**d**) rear view of Mavic2 Enterprise Dual with all onboard sensors.

**Table 1.** Specifications of HWS-19-ONE.

| | |
|---|---|
| size | 15 mm × 38 mm × 7 mm |
| weight | 1 g |
| wind range | 0.0−20.0 m s$^{-1}$ |
| minimum measuring interval | 0.25 s |
| directivity error | ±10% |
| response time | 15 s |
| operating environments | Temperature: 0−50 °C; RH: 20−90% |
| input voltage | 3.5∼5 V |
| data output | serial communication |
| price | <$200 |

### 2.2. A UAV: DJI Mavic2 Enterprise Dual (M2)

The sUAV used in this study was DJI Mavic2 Enterprise Dual (M2), a small quad-copter weighing less than 900 g with a top beacon and bottom light. Its operating temperature is between −10 and 40 °C. Inoue and Sato [15] confirmed that the M2 operated effectively under very cold conditions. The maximum endurance is 31 min. The maximum payload is 200 g, making it challenging to load a commercial ultrasonic anemometer, including a data logger with batteries or a communication device between the M2 and land station.

### 2.3. Device Assembly

A 12-cm length cylinder (15 g) was produced using a 3D printer for mounting the HWS on the M2 (Figure 1d). Based on Inoue and Sato [15], a relative wind direction for the air temperature measurements is critical to reducing the temperature bias caused by the sUAV body and rotors. Therefore, a judgment of whether the headwind or tailwind dominates is essential for high-quality temperature measurement. Accordingly, the HWS was set to the cylinder with 45° deflection (Figure 1b).

On the right front arm, an iMet-XQ2 (International Met Systems, Inc., Grand Rapids, MI, USA) with a radiation shield (total weight: 70 g) was installed to observe air temperature, relative humidity, air pressure, and GPS position with a 1 s interval [15]. The iMet-XQ2 has an independent battery and data logger. The secondary meteorological

sensor (M5Stack, Env II Unit) was installed on the left rear arm with a radiation shield (total weight: 16 g) to measure air temperature, relative humidity, and air pressure. The aerosol number concentration was observed using an SPS-30 (Sensirion: 27 g) at the bottom of the anemometer cylinder on the M2 (Figure 1c).

Except for the iMet-XQ2, all data were recorded every second in the microcomputers (M5 StickCPlus and Atom Lite; M5Stack, Shenzhen, China) stored in a plastic case, including a 3.7 V lipo battery (380 mAh) (refer to the latest assembly state in Discussion). Two small magnets in M5 StickCPlus were removed because they strongly affect the M2's compass.

## 3. Laboratory Experiments

### 3.1. Fan Experiments

Before obtaining the vertical distribution of wind speed by the ascending M2, a bias between a calibrated ultrasonic anemometer (Vaisala WXT536) and HWS was estimated in front of a blowing fan (Figure 2a) at the Kitami Institute of Technology in Japan. The measurements were performed twice over more than 15 min. Table 2 summarizes the results. The HWS has a positive bias ($U'$) of 1.0 m s$^{-1}$ under weak wind speed conditions.

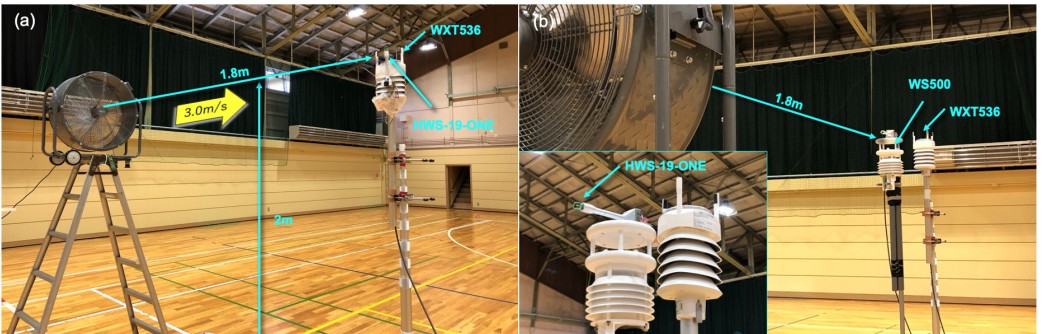

**Figure 2.** (**a**) Horizontal wind experiments, and (**b**) vertical wind experiments.

**Table 2.** Horizontal wind speed test.

|  | HWS (m s$^{-1}$) | WXT536 (m s$^{-1}$) | Bias (m s$^{-1}$) | Averaged Period (min) |
|---|---|---|---|---|
| 1st | 3.9 | 3.0 | 0.9 | 18 |
| 2nd | 4.1 | 3.0 | 1.1 | 15 |

Next, the impact of ascending speed of the M2 (i.e., relative vertical velocity) on the HWS was estimated by tilting the HWS 90° facing the upstream direction from the top of the sensor. Two ultrasonic anemometers (WXT536, Vaisala Oyj, Finland and WS500, OTT HydroMet Fellbach GmbH, Germany) were prepared, and the HWS was installed on the WS500 (Figure 2b). Five ascending speed conditions from 0.8 to 3 m s$^{-1}$ ($W$) were assumed and tested. Table 3 summarizes the results. Note that the observed wind speed ($U_{obs}$) does not contain an actual horizontal wind component under no horizontal wind condition ($U_t = 0$) in this experiment; however, the HWS detected the $U_{obs}$ as a linear function of $W$ with a linear coefficient of 0.625 ($W'$) (Figure 3). The vertical wind bias on the horizontal wind during ascending is expressed as $W' \times W$, where $W$ is the ascending speed of the M2.

**Table 3.** Vertical wind speed test.

| | HWS (m s$^{-1}$) | WXT536 (m s$^{-1}$) | WS500 (m s$^{-1}$) | Averaged Period (min) |
|------|------|------|------|------|
| 1st | 0.47 | 0.85 | 0.76 | 2 |
| 2nd | 0.74 | 1.25 | 1.20 | 2 |
| 3rd | 1.12 | 1.85 | 1.81 | 2 |
| 4th | 1.53 | 2.45 | 2.44 | 2 |
| 5th | 1.88 | 2.80 | 2.98 | 2 |

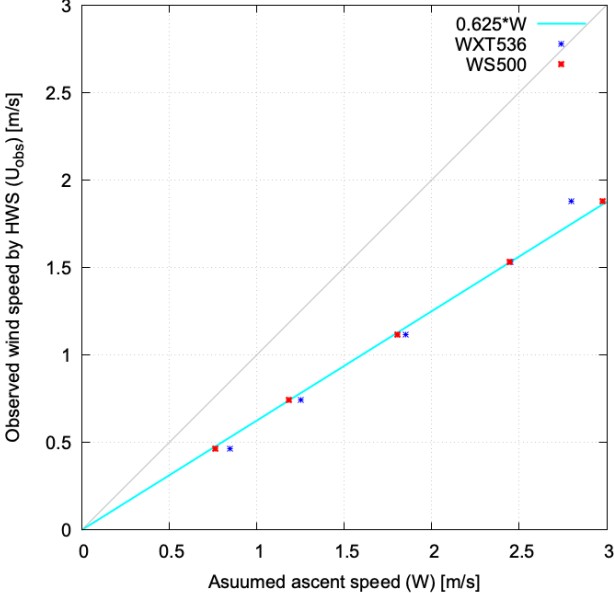

**Figure 3.** Scatter plot of wind speed by ultrasonic anemometers (WXT536 & WS500: *W*) and HWS ($U_{obs}$).

### 3.2. Smoke Visualization Experiments

The wind bias of the flying sUAVs can hardly be estimated because the 3D airflow caused by rotors influences the observed winds. In this study, a smoke experiment was conducted to visualize the airflow of the hovering M2. Two sets of laser light sheet sources with a 532-nm wavelength (Parallel Eye H, Shin Nippon Air Technology Co., Ltd., Tokyo, Japan) visualized the fine particles from two smoke machines (Figure 4a). Images monitored by a professional ultra-high sensitive video camera for fine particle visualization with 600 fps (EYE-SCOPE, Shin Nippon Air Technology Co., Ltd.) were used to estimate particle velocity fields with an image processing package (plus PIV, Shin Nippon Air Technology Co., Ltd.). In this analysis, 1 s of data (600 frames) was used.

The particle movements between two images were calculated for each pair of images (1920 × 1080 pixels; a total of 599 pairs) using the 25 × 25 pixels window. The velocity field was estimated by averaging the results from all pairs. The accuracy of image analysis is ±5% (Mr. Takahashi, personal communication). The hovering altitude was approximately 5 m from the floor. Note that the background wind speed was zero.

Four sets of visualization experiments were conducted by changing the direction of the M2 as follows (Figure 5): exp.1 (facing the front side), exp.2 (facing the left side), exp.3 (facing right front arm to left rear arm), and exp.4 (facing left front arm to right rear arm). In these experiments, the horizontal and vertical velocities at the place of the HWS were focused (magenta arrows in Figure 5a,e). The results are summarized in Table 4. The hovering biases for horizontal wind ($U_0'$) and vertical wind ($W_0'$) were −0.38 (left-rear to right-front) and −1.18 (downward) m s$^{-1}$ on average, respectively. The cause of $U_0'$ is the extra load of the right front rotor from the extra weight of the iMet-XQ2 at the right front arm to maintain the M2's attitude. However, the value of $U_0'$ might be negligible on the

outside because the attitude heavily depends on the horizontal wind speed. Therefore, this study assumes $U_0' = 0$. In contrast, $W_0'$ is not negligible because the vertical component linearly affects the observed value as $W' \times (W + W_0')$.

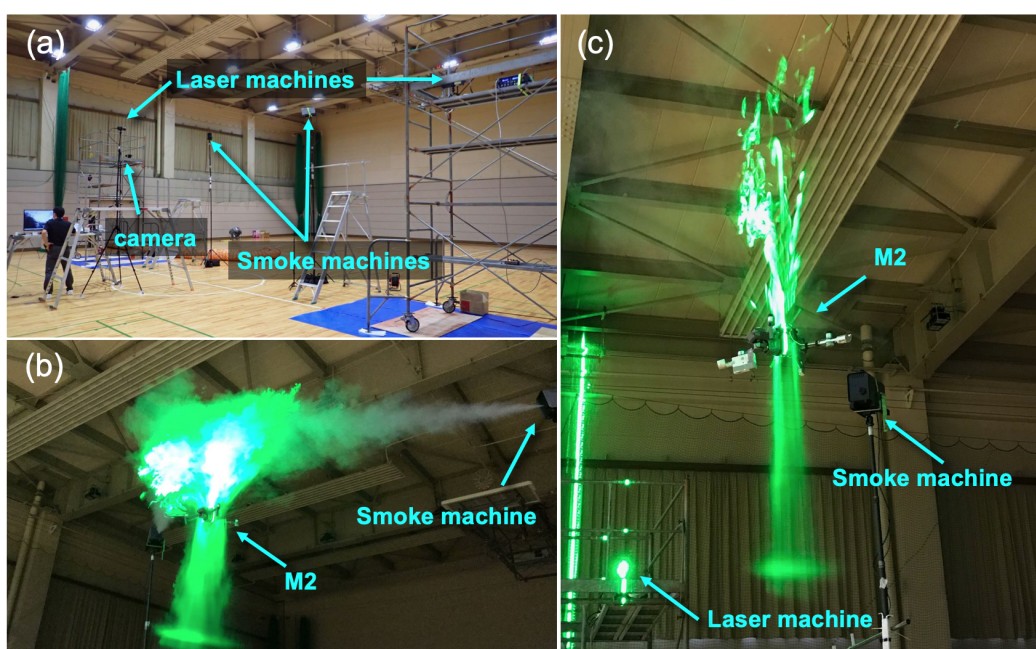

**Figure 4.** (**a**) Setup of smoke experiment, and (**b**) front view and (**c**) side view of the hovering M2 with visualization by a smoke and laser system.

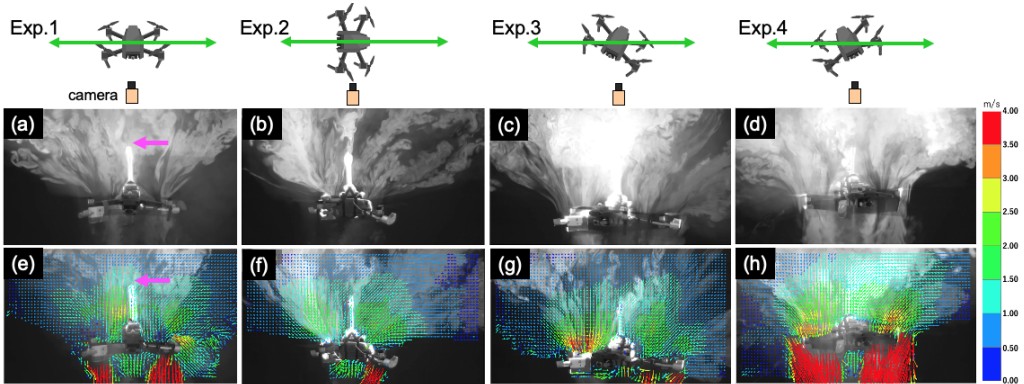

**Figure 5.** Results of smoke experiments. (**a–d**) Raw images and (**e–h**) analyzed velocity fields for each experiment. Magenta arrows indicate the target location where the HWS is installed.

**Table 4.** Smoke experiments.

|         | HWS (m s$^{-1}$) | $U_0'$ (m s$^{-1}$) | $W_0'$ (m s$^{-1}$) |
|---------|------------------|---------------------|---------------------|
| exp.1   | 0.95             | $-0.35$             | $-1.27$             |
| exp.2   | 0.68             | $-0.44$             | $-0.98$             |
| exp.3   | 0.81             | $-0.41$             | $-1.40$             |
| exp.4   | 0.63             | $-0.31$             | $-1.07$             |
| average | 0.77             | $-0.38$             | $-1.18$             |

### 3.3. Bias Correction

The following relationship is considered to estimate the actual horizontal wind speed ($U_t$) from the observed wind speed using the HWS ($U_{obs}$) under the condition of ascending speed of $W$:

$$U_{obs} = \sqrt{(U_t + U' + U'_0)^2 + (W' \times (W + W'_0))^2}$$ (1)

then, $U_t$ is expressed as below.

$$U_t = \sqrt{U_{obs}^2 - (W' \times (W + W'_0))^2} - (U' + U'_0)$$ (2)

The distribution of $U_t$ under the typical ascending motion of the M2 is depicted in Figure 6. Overall, $U_{obs}$ by the HWS on the M2 overestimates by 1.5 m s$^{-1}$ under high wind speed conditions ($>8$ m s$^{-1}$) with a slight difference in the ascent speed ($W$) of the M2. Under the lower wind speed conditions ($<4$ m s$^{-1}$), the HWS overestimated the wind speed by more than 2 m s$^{-1}$, especially under higher $W$, and cannot estimate $U_t$ for $U_{obs} < 2$ m s$^{-1}$.

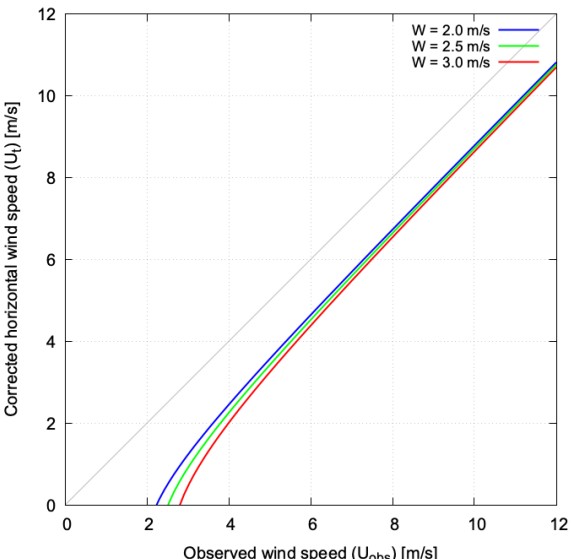

**Figure 6.** Relationships between observed wind speed by the HWS and corrected horizontal wind speed proposed by Equation (2). Each color indicated the case of ascending speed.

## 4. Field Experiments

The HWS on the M2 was applied to obtain the wind profiles over the ice-covered Okhotsk Sea on the Patrol Vessel (PV) *Soya* during 11−15 February 2022. Eighteen profiles were obtained from the ship to 500 m above sea level (ASL, yellow dots in Figure 7a). The ascent speed was approximately 3 m s$^{-1}$. Figure 8a illustrates the time–height cross-section of the corrected wind speed during the cruise. The fine structure of horizontal wind distribution was obtained even in the relatively stronger wind conditions on 14 February ($>7$ m s$^{-1}$).

It is possible to compare the wind data between the ship and the M2 because the same ultrasonic anemometer used in the laboratory experiment was available on the ship (WXT536 at 14-m ASL: Figure 7c). Although the ship wind data were recorded every minute, the stern's tailwind conditions reduced the data's quality because of the modified wind speed and direction caused by the ship mast and body (Figure 7b). Therefore, the ship wind data before the M2 operation within 30 min were compared with the M2 at the same altitude. Five-minute data closest to the M2 operation when the ship moved with a speed of more than 1.0 m s$^{-1}$ were averaged for the analysis.

Figure 8b illustrates the near-surface wind speed time series. As expected, the raw M2 wind speed was usually stronger than the shipboard wind speed. The corrected M2 data were matched relatively closely with the ship wind data. Statistically, the intercept coefficient was reduced from 2.8 (raw) to 0.9 m s$^{-1}$ (corrected), although the correlation

coefficient was at the same level (Figure 9). The root-mean-square error (RMSE) was $\pm 1.13 \text{ m s}^{-1}$, suggesting that the proposed correction method in this study was accurate and that the wind speed was within the WMO OSCAR requirement of significant improvement for high-resolution NWPs (uncertainty of $2 \text{ m s}^{-1}$) [12].

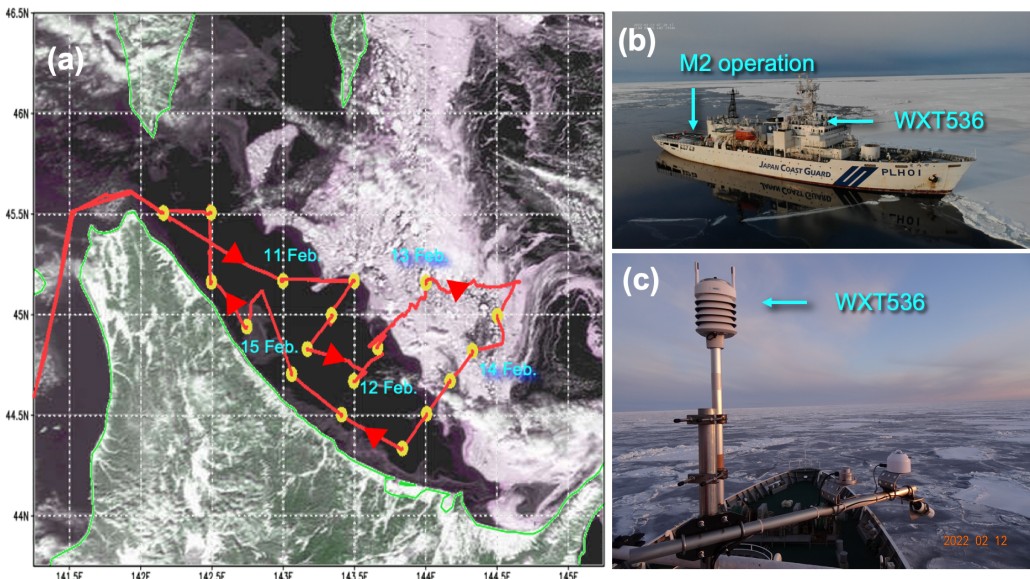

**Figure 7.** (**a**) Cruise map of PV *Soya* with the MODIS image on 10 February 2022 and the M2 profiling stations (yellow dots), (**b**) locations of the M2 operation and the ultrasonic anemometer (WXT536), and (**c**) the closeup image of WXT536 on the upper deck of the ship.

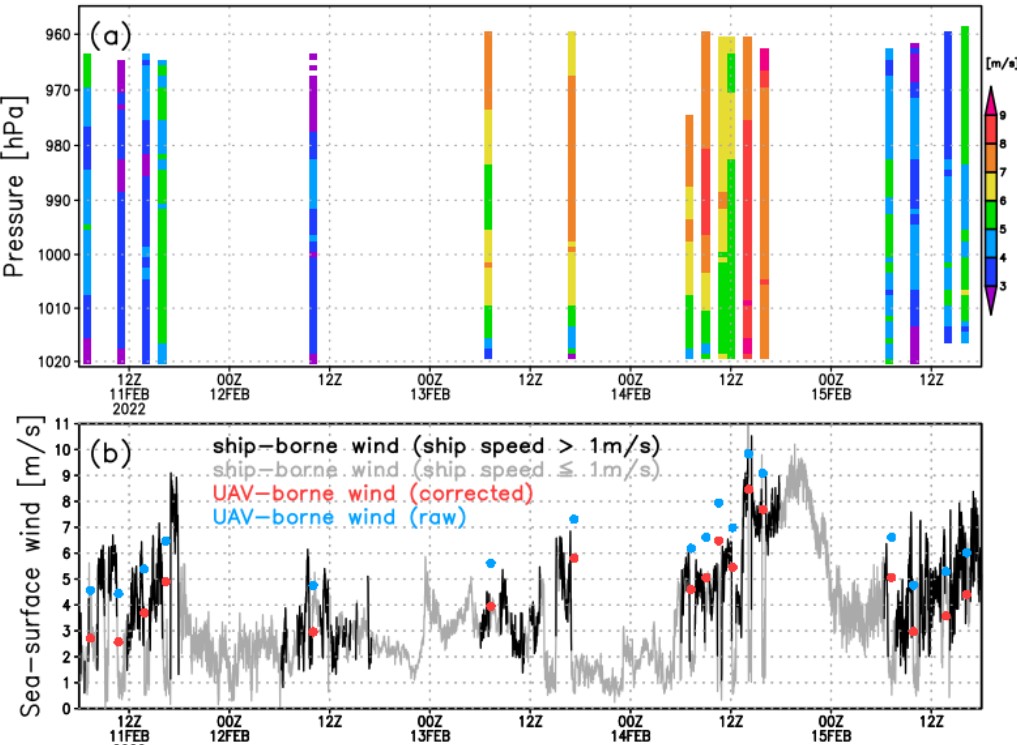

**Figure 8.** (**a**) Time–height cross section of the corrected wind speed obtained by the M2, and (**b**) time series of horizontal wind speed observed by PV *Soya* (black: ship speed > 1 m s$^{-1}$, gray: ship speed ≤ 1 m s$^{-1}$) and by the M2 (blue dots: raw data, red dots: corrected data). The time axis is the local time.

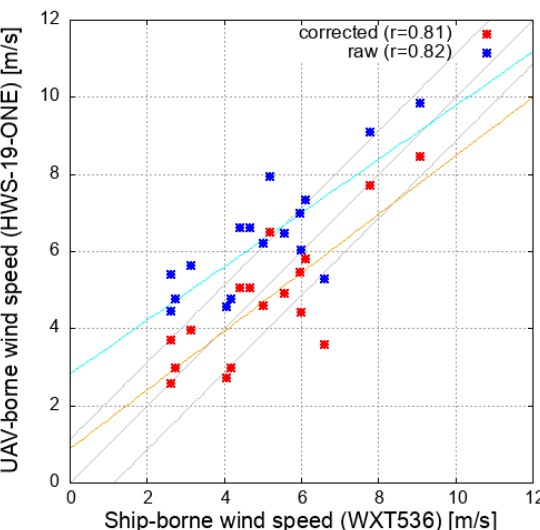

**Figure 9.** Scatter plots of wind speed between PV *Soya* and M2 (red: corrected, blue: raw) with linear regression lines. The values in parentheses are correlation coefficients.

## 5. Discussion

There are many sources of wind speed bias caused by mechanical issues. In this study, only essential components were assessed, i.e., the effect of the ascent motion on the HWS and the airflow feature of the hovering M2. Three-dimensional airflow during the ascent might be another source of bias because the rotors in the upper air (e.g., 1000-m level and higher) must bear the stronger wind speed to maintain the M2 attitude under low air pressure conditions, presumably modifying the airflow. Under strong wind conditions, tilting the M2 and sensor might cause an additional wind speed bias. The mounting height of the sensor (12 cm from the top of the M2) depended on the serial cable length of HWS. The extension of the cable and placement height might reduce the $W_0'$, although the extra payload for the wind sensor cylinder should be considered with a less stable flight attitude. Nevertheless, the corrected wind speed in this study would help analyze the boundary layer structures as a first-order approximation.

Barbieri et al. [18] compared the wind speed among several UAVs relative to meteorological tower data. Three types of ultrasonic anemometers on the hovering sUAVs revealed the most consistent results in minimal deviations from the tower values, although the observed wind speed range was between 1 and 5 m s$^{-1}$. The Coptersonde also provided accurate wind speed by calculating its attitude with an RMSE of 0.77 m s$^{-1}$ under a broader range of wind speeds between 2 and 10 m s$^{-1}$ [20]. However, it was sometimes underestimated under strong wind conditions (>18 m s$^{-1}$) because the wind calculation coefficient was out of range [21].

Inoue and Sato [15] demonstrated that even in an ultrasonic anemometer (Vaisala WXT532), the wind speed of the ascending sUAV (ACSL PF2) has a bias of 1 m s$^{-1}$ compared with radiosondes and that one of the state-of-the-art meteorological sUAVs (Meteomatics Meteodrone MM670) has a wind speed bias of more than 1 m s$^{-1}$. The wind speed measurement with a thermal anemometer is comparable to existing systems (dynamical wind calculations and ultrasonic anemometers).

The estimation of wind direction is one of the challenges for ascending sUAVs because the relative downward airflow disturbs the actual horizontal wind direction when a sonic anemometer is onboard. Inoue and Sato [15] reported considerable wind direction uncertainty ($\pm 90°$) for the onboard Vaisala WXT532, a significant issue regarding how an sUAV performs deep profiling quickly and with high quality in a limited battery capacity.

The HWS can detect the approximate wind direction (whether the headwind or tailwind dominates) using the probe at each side (Figure 1a). With two HWSs, when an HWS is oriented at right angles to the other (Figure 10a), the actual wind direction can be

estimated based on the inverse function of the tangent. The additional benefit of using two sensors enables averaging two wind speed values, reducing the uncertainty of the wind speed. In the latest design, a 920-MHz LoRa antenna board (ES920LR3) was connected to a microcomputer (M5 StickCPlus) to monitor the wind data during the sUAV flight in real-time, contributing to the decision-making of safety operation (Figure 10b).

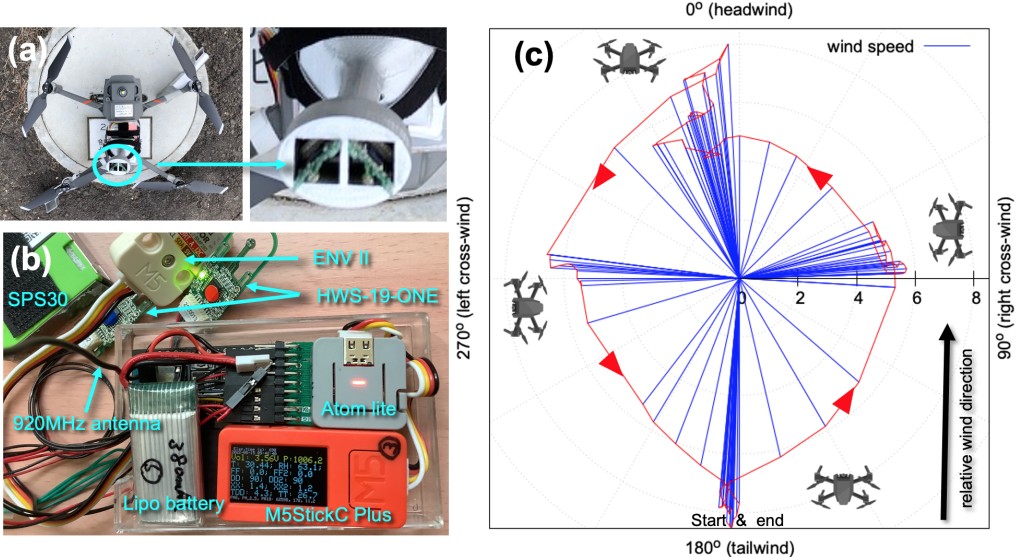

**Figure 10.** (**a**) Dual-mount of HWS, (**b**) assembly state of microcomputers and sensors, and (**c**) rose diagram of wind speed and direction during the hovering test by rotating 360° at 140-m AGL.

On 8 April 2022, the flight test was conducted at the old race track of the Kitami Institute of Technology (Figure 5 in [15]). The M2 flew up to 140-m above ground level (AGL) and stayed under tailwind conditions (180°) by monitoring the wind direction at the land station with 920-MHz communication. Then, the M2 rotated clockwise 90° to catch the right cross-wind. After 15 s, the same rotations were repeated to measure the headwind, left-cross wind, and tailwind.

Figure 10c illustrates the rose diagram of wind speed. The correct change in the M2-relative wind direction was observed, suggesting that the dual-mount of HWS is effective at estimating the wind direction. However, the left and right cross-wind speeds seemed to be underestimated $1-2$ m s$^{-1}$, presumably because the upstream sensor might disturb the airflow at the downstream sensor. The impact of the M2 orientation on the wind speed relative to the wind direction should be assessed in detail in a laboratory experiment.

Although ultrasonic anemometers are not heavy ($<$100 g), extra payloads are expected for the power supply, a mounting pole, and a data storage system, which is suitable for larger sUAVs. There are many issues regarding the uncertainties for wind measurements by thermal anemometers. However, the lightweight (1 g) and low cost ($<$\$200) are substantial advantages to monitoring wind conditions for all sUAVs, especially the small ones with limited payload.

## 6. Conclusions

This study demonstrated the potential benefit of a small and inexpensive thermal anemometer (HWS-19-ONE) for obtaining wind speed profiles using commercial sUAVs. Several biases caused by the sensor shape and the airflow during flight should be considered during the ascent. The laboratory experiments revealed that the sensor has a positive horizontal wind bias ($U' = 1.0$ m s$^{-1}$) compared with the calibrated ultrasonic anemometer and a linear effect of vertical wind that influences the horizontal wind speed ($W' = 0.65 \times W$ m s$^{-1}$) as a function of ascending speed of $W$. The smoke experiments indicated that the airflow during hovering has a descent flow bias ($W_0' = 1.2$ m s$^{-1}$).

During field experiments over the ice-covered Okhotsk Sea, the thermal anemometer system onboard the M2 profiled the wind from near-surface to 500-m ASL. The corrected M2 wind speed, with an RMSE of $\pm 1.13$ m s$^{-1}$, agreed with the shipboard wind speed measured by an ultrasonic anemometer. Finally, the potential capability of the dual-mount thermal anemometer system for estimating the wind direction with real-time data monitoring was proposed, which requires more laboratory experiments.

Wind profiles are essential for operational NWPs and safe UAV operations. Extra payloads for obtaining wind data by an ultrasonic anemometer ($>50$ g) limit the opportunity of the observations, especially for commercial sUAVs. Monitoring wind speed will become more critical for BVLOS flights shortly. The application of inexpensive, lightweight thermal anemometers provides new insights for obtaining the wind data for many available sUAVs.

**Author Contributions:** Conceptualization, J.I; methodology, J.I.; software, J.I.; validation, J.I. and K.S.; formal analysis, J.I.; investigation, J.I. and K.S.; resources, K.S.; data curation, J.I.; writing—original draft preparation, J.I.; writing—review and editing, K.S.; visualization, J.I.; supervision, J.I.; project administration, K.S.; funding acquisition, K.S. All authors have read and agreed to the published version of the manuscript.

**Funding:** This research was funded by The Arctic Challenge for Sustainability II project grant number JPMXD1420318865. The APC was funded by the National Institute of Polar Research, Research Organization of Information and Systems.

**Data Availability Statement:** Field experiment data is available from the Arctic Data archive System (ADS) at https://doi.org/10.17592/001.2022093001 (accessed on 1 October 2022). Other laboratory experiment data may be obtained upon request.

**Acknowledgments:** We are greatly indebted to Jun Nishioka and Takenobu Toyota (Hokkaido University) for providing the opportunity of PV *Soya* cruise. Hyogo helped with the UAV operation. Shin Nippon Air Technologies Co., Ltd. supported the smoke experiments. Hortplan LLC provided the idea to measure wind direction. Philip Pape, from Edanz (https://jp.edanz.com/ac (accessed on 1 October 2022)), edited a draft of this manuscript. Two anonymous reviewers provided valuable comments.

**Conflicts of Interest:** The authors declare no conflict of interest.

## Abbreviations

The following abbreviations are used in this manuscript:

| | |
|---|---|
| BVLOS | beyond visual line of sight |
| NWPs | Numerical Weather Predictions |
| OSCAR | Observing Systems Capability Analysis and Review tool |
| UAVs | Unmanned Aerial Vehicles |
| WMO | World Meteorological Organization |

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
