# Peer review of "Wind Speed Measurement by an Inexpensive and Lightweight Thermal Anemometer on a Small UAV"

_drones, doi:10.3390/drones6100289_

Round 1
Reviewer 1 Report
Dear Editors and Authors,
I would like to send you the review result in this email.
Thank you.
The reviewer.

Author Response
Thank you for your review comments. We appreciate your suggestions.
Below are the replies to your comments.
> There is a shortage of reference articles. Typically, the references should be about 35 to 60 documents for a standard scientific paper.
The quality of the paper does not depend on the number of citations.
This paper is relatively short and focused on a particular field; therefore, the number of citations is relatively small.
> Why did you choose your proposed methods? What are the advantages and disadvantages of your preferred methods?
Due to the limited payload, most small UAVs can not install the ultrasonic anemometer. An inexpensive and lightweight thermal anemometer is suitable for sUAVs, which was addressed in the title of this paper. The disadvantage of this system has been mentioned (i.e., fog, clouds, and precipitation) (P2 line 69); however, such a situation is also unsuitable for sUAVs operation.
> Does the fabric of the sUAV used in this study affect the results of your research?
A hook-and-loop fastener is essential for a safe flight to avoid the drop of sensors.
The authors think that a hook-and-loop fastener does not affect the results.
> What is the limitation of this study? What is the author's following research?
The limitation of this research is that our system was tested only for DJI Mavic2. Other sUAVs made by other companies should be tested. We are conducting other sUAVs installing this anemometer system, which will be forthcoming work.
Reviewer 2 Report
summary:
- two approach to wind sensing on a hovering drone: 1 is based on a sensorless approach but heavy tuning, and 2 is based on additional wind sensing, which often limits size of the drone. The goal is to assess the validity of using a thermal anemometer for wind measurements, which are lightweight and inexpensive.
major concerns:
- how sensitive is the bias to azimuth angle ? This is an experiment that should be done. This point was addressed in the discussion, but seems like it is in the scope of this submission. I would strongly encourage this experiment to be done within the scope of this paper.
- The notation is sometimes confusing: on page 5, line 120 of the text, it is said that Uobs is a linear function of W with a linear coefficient of 0.625 (W'), but later eq(1) used and looks different. Some more explanation is to justify eq(1) and clear definition of the different U and W is needed.
minor comments and suggestions in order of appearance:
- the term "directivity" on line 80 is not defined
- Figure 1c and d should have the wind sensor labeled
- Figure 1b is unclear, I'm not sure what the 45 degree deflection is referring to. Suggest making the figure bigger or make a more close up view of the sensor mounting.
- Figures on page 4 seem out of place with the text, suggest moving to a later page.
- table 2 & 3 has no units, it's not clear to me what these numbers are for
- Table 4: units are not defined.
- Are the smoke experiments also subject to external winds? This was not clear to me from the text.
Author Response
Thank you for your review comments. We appreciate your suggestions.
Below are the replies to your comments.
> how sensitive is the bias to azimuth angle ? This is an experiment that should be done. This point was addressed in the discussion, but seems like it is in the scope of this submission. I would strongly encourage this experiment to be done within the scope of this paper.
The sensitivity of wind speed to azimuth angle (i.e., directivity) examined in detail by the manufacturer (Holtplan, LCC, Japan). A test by changing the wind direction by 10Ëš around the sensor under 5m/s horizontal wind condition revealed that the observed wind speed only varied ±10% (the figure is available at https://www.sg-lab.info/wp_sglab/wp-content/uploads/2021/04/image01-1.jpg). This low directivity indicates that the HWS sensor is a high quality thermal wind sensor.
Above sentences were added in the main text (P2 lines82-87).
> The notation is sometimes confusing: on page 5, line 120 of the text, it is said that Uobs is a linear function of W with a linear coefficient of 0.625 (W'), but later eq(1) used and looks different. Some more explanation is to justify eq(1) and clear definition of the different U and W is needed.
We understand your confusion. The situation in line 120 is the case of laboratory experiment under calm situation with ascending motion. In this situation, Uobs does not contain a true wind speed component (Ut=0) but an apparent horizontal wind speed induced by vertical motion.
The text was added as “Note that the observed wind speed (U_obs) does not contain an actual horizontal wind component ($U_t$=0) under no horizontal wind condition in this experiment; however, the HWS detected the U_obs as a linear function…..”
> the term "directivity" on line 80 is not defined
As replied in the major comment, “directivity” is defined as “the sensitivity of wind speed bias to azimuth angle”. (P2 lines 81-82)
> Figure 1c and d should have the wind sensor labeled
Added the wind sensor labels.
- Figure 1b is unclear, I'm not sure what the 45 degree deflection is referring to. Suggest making the figure bigger or make a more close up view of the sensor mounting.
The zoom up figure was added in Figure 1b.
> Figures on page 4 seem out of place with the text, suggest moving to a later page.
The place was moved to a later page.
> table 2 & 3 has no units, it's not clear to me what these numbers are for
> Table 4: units are not defined.
The unit (m/s) was added in each table.
> Are the smoke experiments also subject to external winds? This was not clear to me from the text.
We added that “Note that the background wind speed was zero.” (P6 line 145)